# Hygromechanical Performance of Polyamide Specimens Made with Fused Filament Fabrication

**DOI:** 10.3390/polym13152401

**Published:** 2021-07-22

**Authors:** Roberto Spina, Bruno Cavalcante

**Affiliations:** 1Dipartimento di Meccanica, Matematica e Management, Politecnico di Bari, 70125 Bari, Italy; bruno.melocavalcante@poliba.it; 2Istituto Nazionale di Fisica Nucleare (INFN)—Sezione di Bari, 70125 Bari, Italy; 3Consiglio Nazionale delle Ricerche—Istituto di Fotonica e Nanotecnologie (CNR-IFN), 70126 Bari, Italy

**Keywords:** rapid prototyping, fused filament fabrication, composite, polyamide, fiberglass

## Abstract

The material performance of polyamide (PA) samples made with fused filament fabrication (FFF) was analyzed. The authors implemented a well-structured framework to identify the filaments main properties before processing them and characterizing the printed samples. Unfilled and glass-fiber reinforced PA were investigated, focusing on moisture absorption and its effects on dimensional stability and mechanical performance. The properties were collected using differential scanning calorimetry and Fourier-transform infrared spectroscopy, whereas the specimens were characterized by employing compression tests. This framework allowed for the moisture determination, as well as the influence of the moisture absorption. A significant impact was detected for the glass-fiber reinforced PA, with a decrease in the dimensional and mechanical performance. The novelty of this study was to define a well-structured framework for testing the moisture influence of FFF components.

## 1. Introduction

The demand for reducing CO_2_ emissions and fuel consumption is continually growing in the USA, Europe, and China. A total of 23% of carbon pollution depends on vehicle emissions during their operation, thus charging the main greenhouse gas (Huang et al. [1]). Regulations require a fuel efficiency improvement of about 5% per year in the USA, whereas a maximum gas emission of 95 CO_2_ per km was set as a target for 2020 and onwards in Europe. Moreover, car manufacturers need to recycle and recover the materials of at least 95% of a vehicle’s weight (European Regulation [2]). As a result of these crucial demands and regulations, innovations in materials, design, and technology are expected to impact the production of cars for the next twenty years. Possible solutions for meeting these environmental goals may include: further improvement of engine capability by reducing the size and number of components, designing and manufacturing structurally efficient vehicle bodies, and cutting the overall vehicle weight (Lyu and Choi [3]). The actual tendency to replace ferrous and non-ferrous metals with lightweight materials, such as polymers and composites, enhances vehicle weight reduction, mechanical properties, and processing efficiency (Akampumuza et al. [4]). Simultaneously, the employment of lightweight materials promotes the development of novel materials and innovative technological processes. However, it is important to be attentive to the fact that said innovations could cause an excessive rise in fabrication costs, which could hinder the feasibility of mass-production (Park et al. [5]). Replacing metal with polymers, or polymer-based composites, is a rising method of fulfilling the above goals and satisfying high load capacity requirements. Several interior and exterior parts are made of polymers with comparable thermo-mechanical properties (and at a lower cost than metals), such as radiators, door beams, driveshafts, and tanks (Zhang et al. [6], Caputo et al. [7], Mouti et al. [8], da Silva et al. [9], and Gratzl et al. [10]).

Focusing on engineering thermoplastics, polyamides (PA) are semi-crystalline polymers with low density and high thermal stability. They are characterized by excellent mechanical properties, valuable wear strength, friction coefficients, temperature, and impact properties. Moreover, their chemical resistance is good, especially in the presence of oils. The reason for their performance lies in the base polymer chemical structure. Additives usually are used to enhance these properties. Njuguna et al. [11] pointed out that the failure strength may be shortened, due to the presence of high stress concentration areas. The addition of glass fibers has several positive effects, such as increased stiffness and heat distortion temperatures, while significantly reducing mold shrinkage and dimensional instability. However, moisture lowered the performance of the PA parts. Several authors have studied the relationship between moisture and material properties. Kohan [12] mentioned that the propensity to absorb moisture limits the extensive use of PA. The moisture negatively influences product performance in terms of processability, dimensional stability, and mechanical and chemical properties. Kim et al. [13] pointed out that property degradation occurs at temperatures above the glass transition. The water absorption has a decisive impact on components exposed to various weather conditions during service. For this reason, it is essential to select materials with deficient moisture absorption, or design components to prevent it. Ksouri et al. [14] described that the macroscopic properties of most PA were influenced by hygrothermal aging, limiting the use of these materials. Different phenomena may appear (decreasing the component efficiency and durability), such as breaking the pre-existent bonds caused by water, with a loss of mechanical cohesion, as well as the degradation of the interface between the fibers and matrix. The process, and its additional operations, may alter the moisture content. Claveria et al. [15] illustrated the influence of moisture on the injection-molded parts. The raw materials are generally dried to eliminate moisture, preventing aesthetic and mechanical defects during and after the process. Parts without moisture at the mold injection absorb it from the environment to reach a new equilibrium condition, with a variation in part dimensions. Valino et al. [16] compared injection-molded parts with additive ones. The fabrication speed of injection-molding is higher. However, essential time savings are possible, due to the absence of the mold and its fabrication. Moreover, the high viscosity and random fiber distribution may affect the final quality of the injection-molded components. Van de Werken et al. [17] underlined that the additive manufacturing processes are exciting alternatives to realize parts with reinforced polymers, thanks to their ability to extrude polymers in filament and then deposit them in designed paths by giving directional properties. Polymers may be reinforced with short or long fibers, depending on chosen machines and processes. Miguel et al. [18] suggested conducting water absorption tests on additive parts to evaluate improvements when using a coating to reduce the process-induced porosity. The strength and stiffness of the resulting coated parts are different from untreated ones, but acceptable for non-load bearing applications. Continuous developments in additive manufacturing improve process capabilities and velocity, expanding the material ranges and applications. Suarez and Dominguez [19] pointed out the more favorable environmental balance of additive processes for their near-shape fabrication with reduced wastes. Another improvement should be addressed to reduce energy consumption, improve part constancy and efficiency, and to promote material recycling, to cite a few. The energy consumption data are typically not available to compare the two classes of processes, due to their significant difference.

In this study, the material performance of PA samples made with fused filament fabrication was analyzed. This required a well-structured framework to identify the main properties of the filaments, before processing them and characterizing the samples. Unfilled and glass-fiber reinforced PA polymers were investigated, focusing on moisture absorption and its effects on dimensional stability and mechanical performance. The material properties were collected using differential scanning calorimetry and Fourier-transform infrared spectroscopy, whereas the printed specimens were characterized by employing the compression tests.

## 2. Materials and Methods

The experiments were performed on PA filaments, identifying the thermal properties before processing, and then evaluating the sample’s mechanical performance.

### 2.1. Materials

The investigated materials were two PA filaments, called TECHLine and xStrand, the diameters of which were 1.75 mm. The TECHLine (TAG in 3D, Arnas, France) was a natural-colored PA6, with a waterproofing skin. The xStrand (Owens Corning, Toledo, OH, USA), was a black-colored PA6, with 30% short-fiber glass reinforcement. Both PA materials were ideals to produce functional prototypes and complex parts, thanks to their high strength, wear resistance, and good adhesion to the build plan. Table 1 reports the main physical and mechanical properties declared by the suppliers.

### 2.2. Material Characterization

This activity was divided into several steps: to characterize the materials, realize the samples, perform aging and compression tests, and, finally, analyze results. The entire process is represented in Figure 1. The materials in the dry state follow the path 1-2-3-6-7, whereas the conditioned and saturated specimens followed all steps of the path. The choice of a conditioned state was defined after the identification of the saturated state.

### 2.3. Filament Characterization

The primary characterization analyses of the filaments were carried out via differential scanning calorimetry (DSC) and Fourier-transform infrared spectroscopy (FT-IR). The DSC tests were functional for identifying the thermal temperature profile for single temperature peaks. The sample was put into an aluminum pan, and then sealed to minimize its interaction with the atmosphere and inhibit the evaporation of the more volatile materials studied. An empty pan, with a cover, was used as a reference. The crucibles were weighed before and after each experiment to evaluate possible changes in mass. The measurements were carried out on a Chip-DSC 10 (Linseis Messgeräte GmbH, Selb, Germany) and a heat-flux DSC with an integrated heater and temperature sensor. The amount of absorbed and released energy was measured by the sample subjected to a controlled temperature program. The thermal cycle consisted of two sequential runs, with a heating step from 0 to 220 °C, a holding step for 5 min, and a cooling step from 220 to 20 °C, for samples weighing about 6 mg, according to a well-known procedure (Spina [20]). The heating and cooling rates of the imposed thermal cycle varied between 25 and 100 K/min.

In the FT-IR analysis, the polymer sample, cut from the filament, was placed in the instrument to acquire its spectrum and identify the chemical nature. Samples were placed on the ATR crystal, and the ATR-FT-IR spectrum was scanned after applying pressure. This analysis is the simplest and most sensitive method to analyze microstructure, stereoregularity, branching, or cross-linking. This identification was crucial to relate the structural features to the performance of the polymer (Charles et al. [21]). In this study, a FT-IR spectrometer, ALPHA II (Bruker, Leipzig, Germany), was employed to identify the basic structural units in the chemical configuration. Spectra were collected between 400 and 4000 cm^−1^, with a resolution of 4 cm^−1^, performing 32 scans per spectrum.

### 2.4. Filament Drying

The filament spools were dried using a TR60 digital PID-controlled oven (Nabertherm, Bremen, Germany), with a maximum working temperature of 300 °C and forced air circulation. The target temperature of the oven was set above the glass transition (80 °C) and kept for 8 h to remove all moisture from the materials. The spools were placed in the oven, after the target temperature was reached, to avoid possible overshoot during heating, which could cause plastic softening or a partial fusion. The filament spools were then removed from the oven before use and placed inside the machine enclosure with desiccant to preserve a low moisture level. The FFF machine kept the filaments at a set temperature using a forced air-circulation system in the spool chamber.

### 2.5. Specimen Fabrication

The print tests were carried out using a Funmat HT (Intamsys Technology Co. Ltd., Shanghai, China), a commercial FFF machine used to realize all samples in this experiment. The apparatus has a single extrusion nozzle, mounted on a Cartesian axis system controlled by stepper motors. The 1.75-mm polymer filaments were mounted on a spool, processed through a gear and pinch wheel system into a heated stainless-steel nozzle of 0.5 mm-diameter, and extruded onto the heated platform. The PA6 specimens required a layer of polyvinylpyrrolidone (PVP) adhesive glue on the borosilicate glass plate to ensure good adhesion of the first layer during printing. The PA6GF perfectly adhered to the plate from the first layer and throughout the deposition of the extrudate, without the aid of adhesives. The geometrical shape of the samples was cylindrical, with dimensions of 20-mm diameter by 20-mm height. These dimensions were chosen for determining Young’s modulus, according to ISO 604:2008—“*Determination of compressive properties of plastics*”. The results of this test included compressive strength, compressive yield strength, offset yield strength, and modulus of elasticity. The Cura software (Ultimaker, Utrecht, The Netherlands) was used to slice the CAD model of the part and compute the deposition path, as Figure 2 shows.

The processing parameters used to realize the samples on the FFF machine are reported in Table 2, paying particular attention to the choice of correct extrusion, bed, and chamber temperatures. These temperatures were selected after analyzing the DSC results, in terms of melting and glass transition temperatures. The infill density was set to 100%, via a line infill pattern with an alternative angle of ±45°. The choice of a 100% infill was fundamental in achieving the highest value for the Young’s Modulus, avoiding the difference induced by selecting the infill pattern. A raft adhesion scheme ensured the better sticking of the sample to the build plate.

### 2.6. Specimen Conditioning

The samples were measured immediately after fabrication, considering them in a dry-as-molded state. Average heights and diameters were measured with a digital caliper (accuracy of 0.02 mm). Weights were determined using a LE225D analytical balance (Sartorius AG, Göttingen, Germany), with an accuracy of 0.01 mg for a maximum weight capacity of 110 g. The balance was equipped with the YDK01 density determination kit, based on the Archimedean principle for determining the specific gravity of a solid.

The density 𝜌*_eff_*, using the corrected formula, was computed as:(1)$$\rho_{eff}=\frac{W_{a}\times \left(\rho_{l}-\rho_{a}\right)}{0.99983\times \left(W_{a}-W_{l}\right)}+\rho_{a}$$

*W_a_* and *W_l_* are the weight in air and liquid, while *ρ_l_* and *ρ_a_* the density of the liquid and air. Correction factors were used to consider the influence of the air buoyancy and the depth of immersion.

The density test assessed the deposition performance of the FFF machine. Measurements were repeated for specimens in a saturated state. The absorption rate in a standard atmosphere 23/50 (air temperature of 23 °C, with a relative humidity of 50%), as specified by ISO 291:2008—“*Plastics*—*Standard atmospheres for conditioning and testing*”, required a very long time to reach the level of humidity at equilibrium. A conditioned atmosphere was used instead, to speed up the analysis by increasing the temperature and relative humidity. The method, specified by the ISO 62:2008—“*Plastics*—*Determination of water absorption*”, allowed the amount of moisture absorbed by the specimens immersed in a bath under controlled conditions to be determined. The bath consisted of water with 10 g/L of sodium carbonate (Na_2_CO_3_) and 5 g/L of a surfactant, to lower the surface tension and promote wettability. The sample was initially put in an oven at 50 ± 2 °C for 24 h, extracted from the oven, and weighted to measure the mass value *m*_1_, restoring the dry-as-mold state. The sample was later immersed in a vessel with boiling water for 30 ± 2 min, extracted, and cooled in the conditioning bath at room temperature for 15 ± 1 min. After removing all excess liquid, weighing was carried out within 1 min from the bath removal, registering the mass value *m*_2_. The moisture content (*c*) was identified using:(2)$$c=\frac{m_{2}-m_{1}}{m_{1}}\times 100\;$$

The measuring procedure was repeated at intervals of 30 ± 1 min, until reaching saturation, achieving a complete conditioning test in 1 h.

### 2.7. Compression Test

Uniaxial compression tests were executed at room temperature to study the static behavior of the samples as a function of the moisture content. A servo-controlled 4485 machine (Instron, Norwood, MA, USA) was employed, with load and displacement accuracies equal to 0.25% with the 200 kN load cell, and 2.5 × 10^−5^ mm, respectively. The initial deformation speed was 8.33 × 10^−4^ 1/s, setting a crosshead speed of 1.0 mm/min and a 0.5 N preload before each test, according to the specifications of ISO 604:2002—“*Plastics*—*Determination of compressive properties*”. The test was performed by reducing the height from 20 to 8 mm, achieving a maximum deformation equal to 40%. Teflon discs (Ø50 mm and h = 5 mm) were placed between the planar surfaces of the sample and machine supports to reduce friction, avoiding excessive surface barreling.

### 2.8. Image Analysis

Optical microscopy allowed the structure of the 3D printed samples to be examined. The samples were mounted using cold cure epoxy resin, mixing the resin and hardener at a ratio of 2:1, and cured at room temperature for at least 12 h. Samples were ground using 220, 1000 e 4000 SiC papers, and polished using 9 μm and 3 μm monocrystalline diamond suspensions on polishing cloths. Final polishing was then realized with a 0.05 μm alumina suspension. The multi-purpose zoom microscope system AZ100M (Nikon Instruments Europe BV, Amsterdam, The Netherlands), with a high-resolution camera and a variable magnification lens, was mounted and used to acquire the images of the sample sections.

## 3. Results and Discussion

The DSC analysis repeated the thermal cycles three times, with different samples, and averaged the results. Endothermic and exothermic heat flows were recorded as a function of temperature. Several calibrations were carried out at different rates, with various melting point standards (indium, zinc, and lead). Calibration with a specific rate could not ensure result consistency with other rates. Figure 3 and Figure 4 show the DSC measurements for heating and cooling rates, ranging from 25 to 100 K/min. The trend for different rates could be achieved using (Spekowius et al. [22]). The averaged results are reported in Table 3.

The melting temperature (*T_m_*) of both materials changed with the heating rate; it shifted towards higher temperatures with a rate increase. Moreover, a rate decrease caused a decrease in the melting enthalpy Δ*H_m_*. The same occurred with the glass transition temperature (*T_g_*). As for the crystallization temperature (*T_Cr_*) and enthalpy (Δ*H_Cr_*), they shifted to lower values with higher cooling rates. These results agreed with the literature (Parodi et al. [23] and Pesetskii et al. [24]).

The outcome of the DSC analysis pointed out that the PA6 was less sensitive to the rate variations than PA6GF, and more energy was necessary to melt the filament during processing. In addition, high cooling rates promoted the amorph state. Moreover, these results confirmed the correct choice of the process parameters, selected in Table 2, regarding extrusion and bed temperature, respectively, above the melting and transition temperatures for both materials. The extrusion temperature was one of the most critical settings, because a low value caused extrusion problems, such as poor layer adhesion or a clogged nozzle. On the contrary, a high temperature led to filament burning and nozzle/extruder clogging. The bed temperature selection was essential to avoid a fast raster contraction after deposition caused by rapid cool down.

The FT-IR spectra are shown in Figure 5. The band 3274–3080 cm^−1^ corresponded to the stretching vibration of the amide (N–H) groups, both with and without hydrogen bond interactions with the C–O groups. The bands at 2920 and 2853 cm^−1^ were associated with the asymmetric and symmetric stretching of the methylene (CH_2_) groups. The bands at 1635 and 1538 cm^−1^ were assigned to the stretching vibration of C=O (amide I) and C–N (amide II, CO–N–H bending vibration). The band at 1256 cm^−1^ coincided with C–N stretching (amide III, C–H in-plane bending vibration). The band at 928 cm^−1^ was associated with C–CO stretching in the crystalline α- and β-phases, whereas the band at 1125 cm^−1^ was attributed to the amorphous phase. The presence of the glass fibers was associated with the Si–O–Si band measured between 1200 and 900 cm^−1^. The results confirmed that the materials were very similar, with a difference in the lower bands due to the glass fibers. The analysis of FT-IR was essential to assess the influence of the oxygen effect at high temperatures on filaments. The FT-IR spectra after deposition did not change, concerning those before processing. In this way, the chemical stability of both materials, in regard to oxygen, was evaluated and the conditioning results, achieved with manufactured specimens, were not affected by the environmental conditions.

The fabrication of the compression samples was then performed. Table 4 and Table 5 report the values of the height (*H*) and diameter (*D*) of each specimen, averaged after several measurements, at different diameters and heights, as well as the weight (*m*_1_). The volume (*V*), density (𝜌), and error (*e*) were computed. The error (*e*) was the difference between the density specified in the supplier material datasheet and the measured density. A sample made with the filament deposition presented a higher porosity than a part realized with the traditional fabrication processes, such as injection-molding or machining. The fabrication process with the FFF machine introduced minor geometrical deviations. Measurements were conducted in a dry state for all specimens. Samples made with PA6 had a diameter and height equal to 19.96 ± 0.05 mm and 19.90 ± 0.06 mm, whereas samples made with PA6GF had a diameter and height equal to 19.96 ± 0.05 mm and 19.82 ± 0.09 mm. The higher reduction in diameter for the PA6GF could be associated with the presence, and random distribution, of the glass fibers. The decrease in volume was very low, with values of 6206 ± 40 mm^3^ and 6157 ± 41 mm^3^ for PA6 and PA6GF.

The density (𝜌), compared with the nominal density (𝜌*_nom_*) declared by the suppliers, allowed for the evaluation of the average error (*e*). This error, equal to 0.39 ± 0.28% and 0.47 ± 0.36% for PA6 and PA6GF, did not influence the results of the subsequent tests.

The conditioning procedure started. The first three specimens (*P1*–*P3*) were kept at a dry state, the second three (*P1c*–*P3c*) were put in a controlled state, and the last three (*P1s*–*P3s*) reached the saturated state. It was essential to later evaluate the differences among these states and the mechanical performance of the specimens, as a function of the moisture content. Particular attention was paid to determine the saturated condition and, consequently, choose the controlled state. Applying the procedure described for the specimen conditioning, the moisture content (*c*) of the *P1s*–*P3s* samples was measured every hour, until saturation was attained. The moisture content increased with the immersion time. After five cycles, the complete saturation was reached, obtaining values of 2.37 ± 0.56% and 10.52 ± 0.40% for PA6 and PA6GF.

Three stages characterized the process, as Figure 6 shows, such as (a) rapid absorption with a linear increase, (b) reduction of the absorption rate, and (c) saturation stage with a constant moisture absorption value.

Two cycles were sufficient for conditioning the *P1c*–*P3c* samples, recording values of 2.02 ± 0.59% and 6.82 ± 0.34% for PA6 and PA6GF. The high moisture content absorbed at saturation demonstrated the highly hygroscopic character of the polyamide matrix of PA6GF, despite the presence of a 30% glass fiber reinforcement. The moisture was mainly adsorbed from the matrix, due to the almost nonexistent moisture-absorption property of glass fibers. The presence of the skin treatment on PA6 allowed maintaining a low value of moister absorption. The reason for PA absorption behavior was associated with the water molecules bound by hydrogen bonds between amides, increasing the chain segment mobility, due to the enlargement of the molecular chain spacing (Sambale et al. [25]). The effect of moisture absorption caused an increase in sample size.

The diameter (Δ*D*) and height (Δ*H*) variations are reported in Figure 7, from the dry to saturation state, computed with:(3)$$\Delta D=\frac{D_{state}-D_{dry}}{D_{dry}}$$
(4)$$\Delta H=\frac{H_{state}-H_{dry}}{H_{dry}}$$

*D_dry_*, *H_dry_*, *D_state_*, and *H_state_* were the diameter and height values in mold and investigated state (conditioned or saturated). The diameter variation (Δ*D*) was moderate for both materials, with an average increase of less than 0.5 ± 0.1%. The situation changed for the height variation (Δ*H*), because the measured values were 0.9 ± 0.08% for PA6 and 2.4 ± 0.14% for PA6GF. The higher value of PA6GF was decidedly more significant, considering the presence of the glass fibers, and confirmed the high hygroscopic nature of the PA matrix.

The results of the compression tests should confirm that a progressive increase in the moisture content affected dimensional variations, deformation, and rupture mechanisms. Figure 8 shows the results on dry state, allowing for the comparison of the stress-strain curves and the final deformation states of both materials. The figure also reports the images of the specimens at the end of the test. The PA6 samples were characterized by a reduced elastic range and a pronounced barreling effect, due to a more excellent material ductility. The PA6GF was more fragile and generally fractured at the end of the test, because the glass fibers acted as brittle materials. The stress-deformation curve of PA6 samples was like a ductile material.

The linear elastic zone was followed by an intermediate plastic plateau, with a constant stress value, and via a final rising ramp, stopped at the imposed value of 40% of deformation. The material acted ductile without a break. On the contrary, the PA6GF samples had a non-linear hardening behavior, typical of a fragile material. Not all sections concurrently reached the yield condition, because of inhomogeneous properties at a micro-structural level due to the glass fibers. The failure initially started in a specific area of the sample. The transmitted load through the surrounding sections diminished from failure.

These areas began to unload elastically to restore the stress level, and the deformation localized in a limited sample area. Therefore, the total load decreased, showing a descending trend in the curve after the maximum stress value. The main parameters used to compare the two materials were the compressive modulus (*E_c_*), the stress (*σ_y_*) at which an increase in strain occurred (without an increase in stress), the stress (*σ*_10_) at which the deformation was equal to 10%, and the maximum compressive stress (*σ_m_*). The stress *σ_y_* and *σ*_10_ were equivalent, considering the different behaviors of the two materials. Table 6 reports the average results extracted from the stress-strain curves. The average *E_c_*, *σ_10_*/*σ_y_*, and *σ_m_* values were 795.34 ± 28.80 MPa, 77.24 ± 2.80 MPa, and 92.46 ± 0.75 MPa. The same properties for the PA6GF were 835.99 ± 22.38 MPa, 72.91 ± 4.93 MPa, and 145.35 ± 1.44 MPa. These variations were limited, showing the specific stability of the materials in a dry state.

The effects of the moisture became evident for the conditioned and saturated samples (see Figure 9). The PA maintained an ideal stress-strain behavior, with a *σ_y_* of 70.30 MPa and a *E_c_* of 772.35 MPa. These values confirmed that a slight increase in moisture absorption (2.02%) produced a slight decrease in strength performances, due to interactions between the polymer chain and water molecules. The variation of *σ_m_* was limited, registering 888.56 MPa. The PA6GF seemed to be strongly affected by the conditioning process. The moisture content of 6.75% led to a decrease in *σ*_10_ to 48.60 MPa, with an evident increase in ductility, due to a great reduction in the *E_c_*, to 482.90 MPa. A significant reduction was also recorded in *σ_m_*, to a value of 108.12 MPa. Moreover, the sample structure was also irremediably compromised, showing apparent flaking of the sample shell, due to the weakening and interruption of the intermolecular polymer chains promoted by the action of the water. The saturation state determined minimal variations in the mechanical performances of the PA specimens. The maximum reduction on the *E_c_* was equal to 5.13% for the sample in the saturated state. On the contrary, the PA6GF specimens were strongly affected by the high hygroscopicity of the PA6 matrix. The compression test results were a *σ*_10_ of 25.44 MPa, *σ_m_* of 55.71 MPa, and a *E_c_* equal to 261.35 MPa. The maximum reduction on the *E_c_* was equal to 68.74% for the sample in the saturated state. All results are summarized in Table 7. 

The images of the samples (Figure 10) at the end of the test for the three conditions showed how the PA6 samples maintained their integrity, with an increase in the moisture content. On the contrary, the PA6GF pieces were decisively damaged, with an increase in the moisture content and the brittle nature of the glass fibers. These phenomena led to the introduction of voids in the structure, with fast crack development in the sample, thereby reducing material performance.

The images of the specimen sections had sufficient quality for the qualitative analysis (Figure 11). The section of the PA6 specimens revealed that the deposition rods were homogeneous, leading to a uniform structure with few voids. Also, debris collected in the larger voids of the sample (most notably in the infill regions) contributed to the scratches, as shown in the images. The infill pattern could not be identified for this material. On the contrary, the deposition rods and the presence of the carbon fibers were more evident in the PA6GF section. The short carbon fibers dispersed in the PA6 matrix were highly oriented with the printing directions. The structure was not uniform, with several voids and discontinuities among rods. This inconsistent structure could justify reduced compression strength and increase the moisture content during the conditioning of the PA6GF specimens (concerning the PA6 ones).

## 4. Conclusions

The effect of moisture content on the mechanical properties of unfilled and glass fiber reinforced PA6 was studied. The moisture content increased with immersion time in a controlled bath, reaching the saturation state 5 h after the first immersion. The moisture absorption caused an increase in the dimensional variation of the printed specimens. Moreover, the results showed limited effects on the mechanical properties of the unfilled PA6, with a reduction of about 5%. A significant impact was instead detected for the PA6GF, with a decrease of the dimensional and mechanical performance in the saturated state. Moreover, the integrity of the specimens at the end of the mechanical test was compromised, with severe cracks on the external and internal surfaces. The performance degradation of the PA6GF samples was 33% (yield stress) and 65% (maximum stress). This behavior was consistent with reducing the compressive modulus of the PA6GF specimens by 68% in the saturated state. The drops were ascribed to a decisive degradation of the bonding between the glass fibers and PA matrix, due to moisture. Further research will be addressed to vary the infill percentage and pattern, evaluating the effects on moisture content.

## Figures and Tables

**Figure 1 polymers-13-02401-f001:**
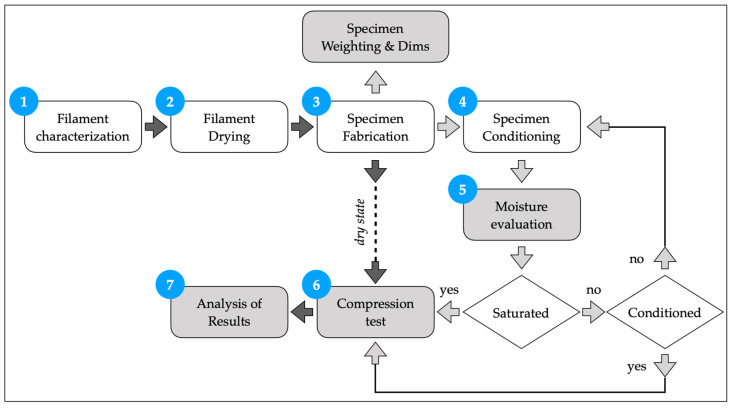
Flow diagram of the experimental activity.

**Figure 2 polymers-13-02401-f002:**
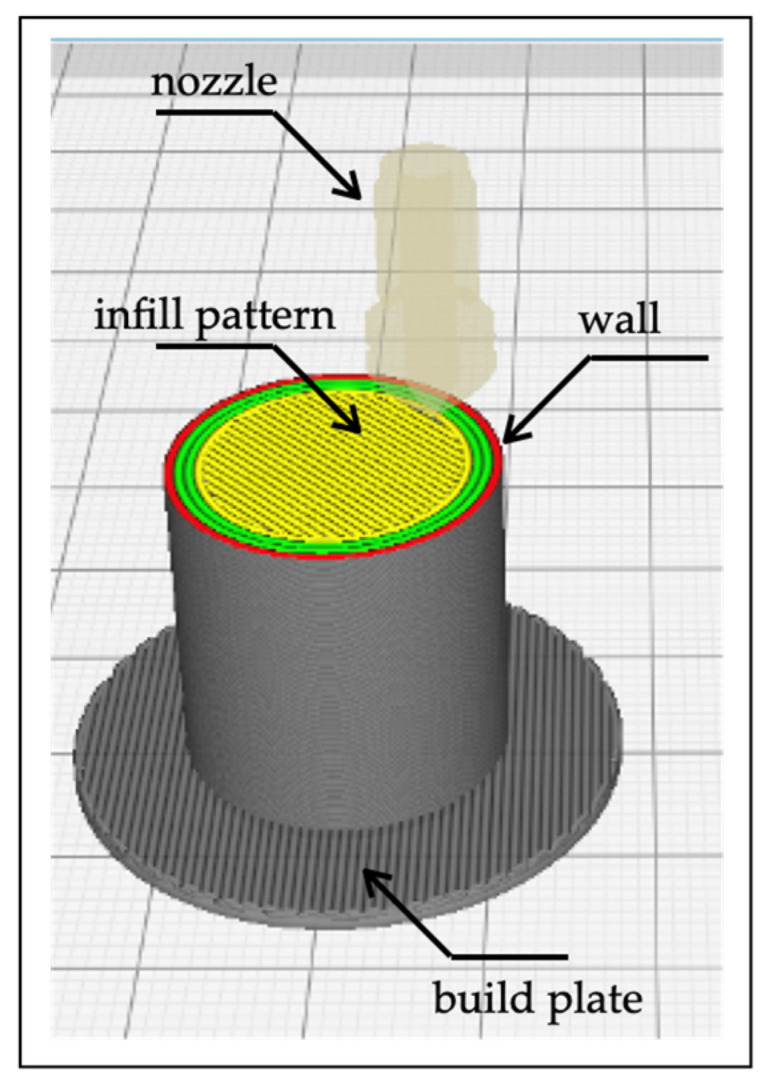
Specimen fabrication.

**Figure 3 polymers-13-02401-f003:**
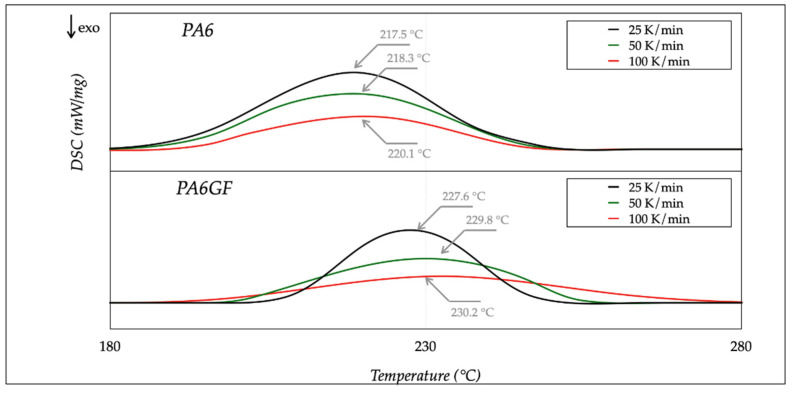
DSC thermograms: heating steps at different rates.

**Figure 4 polymers-13-02401-f004:**
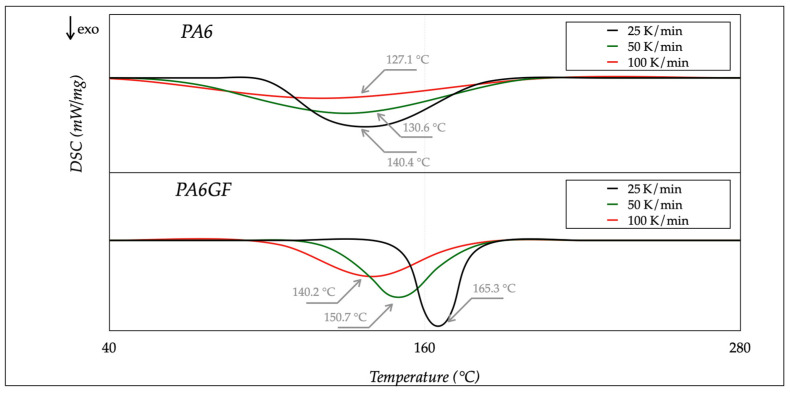
DSC thermograms: cooling steps at different rates.

**Figure 5 polymers-13-02401-f005:**
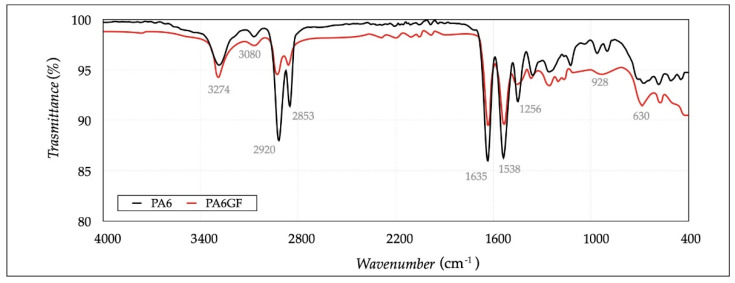
FT-IR spectra of the two materials.

**Figure 6 polymers-13-02401-f006:**
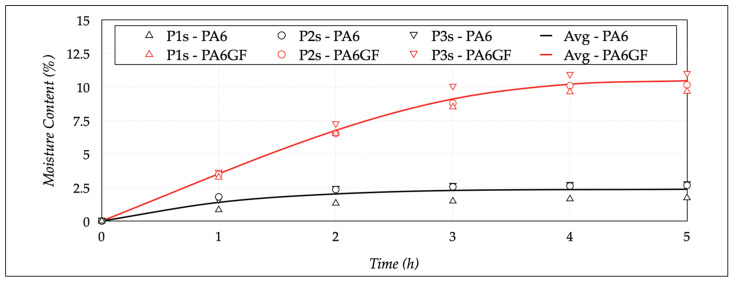
Results of conditioning and saturation tests with time.

**Figure 7 polymers-13-02401-f007:**
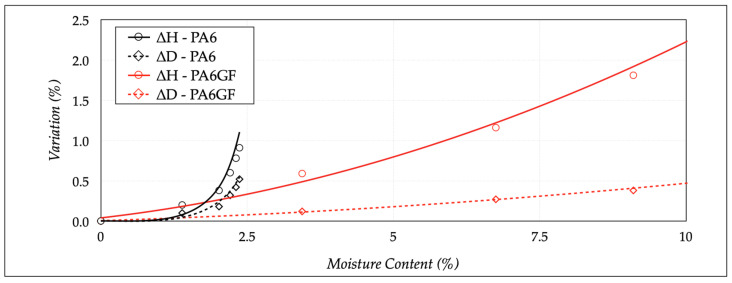
Dimensional variations with moisture content.

**Figure 8 polymers-13-02401-f008:**
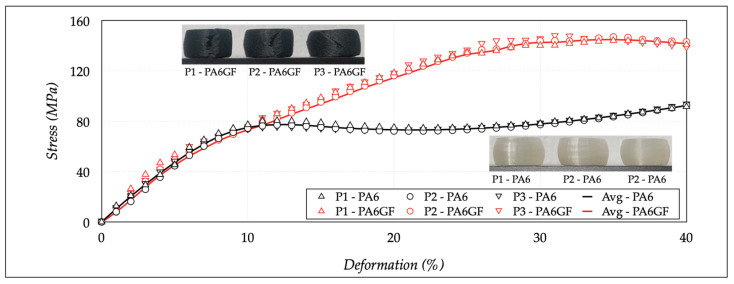
Compression tests for dry-as-mold specimens.

**Figure 9 polymers-13-02401-f009:**
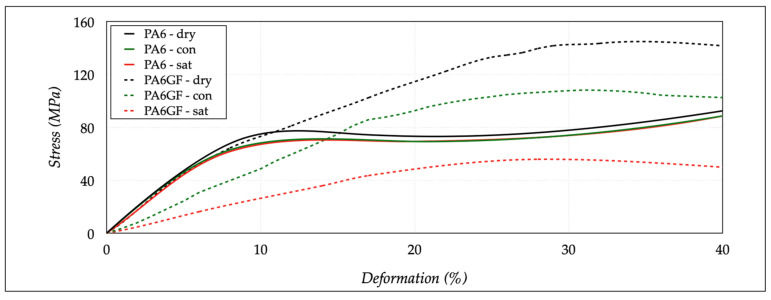
Stress-strain curves of dry, conditioned (con), and saturated (sat) specimens.

**Figure 10 polymers-13-02401-f010:**
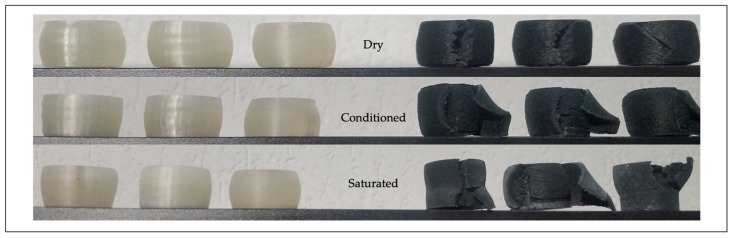
Dry, conditioned, and saturated specimens at the end of the tests.

**Figure 11 polymers-13-02401-f011:**
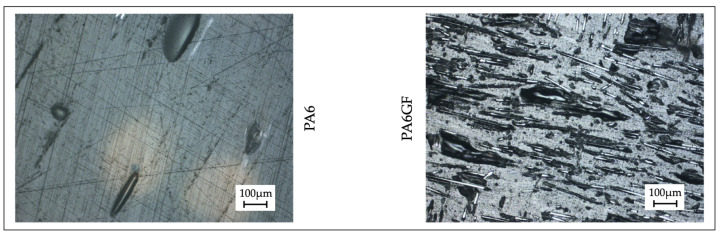
Sections of the printed samples (PA6 on the left side and PA6GF on the right side).

**Table 1 polymers-13-02401-t001:** Main properties of the investigated PA6 materials.

	PA6 (TECHLine)	PA6GF (xStrand)	
Property	Value		Unit
**General**			
Material class	PA6	PA6	–
Reinforcement (glass fiber)	0	30	%
**Physical**			
Density	1.14	1.17	g/cm^3^
Glass transition temperature	49	62	°C
Melting (Softening) temperature	195	206	°C
**Mechanical (Tensile test)**			
Young’s modulus	2400	740	MPa
Yield tensile strength	61	102	MPa
Ultimate tensile strength	82	102	MPa
Elongation at break	9.6	2.1	%

**Table 2 polymers-13-02401-t002:** Process parameters adopted for the FFF specimen fabrication.

Property	Value	Unit	Property	Value	Unit
**Shell and quality**					
Layer height	0.1	mm	Top layers	8	-
Wall thickness	0.8	mm	Bottom layers	8	-
**Infill**					
Infill density	100	%	Infill pattern	lines	-
**Print and support**					
Extrusion temperature	270	°C	Chamber temperature	80	°C
Bed temperature	80	°C	Build plate adhesion	raft	-
Print speed	45	mm/s	Support speed	30	mm/s

**Table 3 polymers-13-02401-t003:** Thermal properties of PA6 materials.

		PA6			PA6GF		
Property	25 K/min	50 K/min	100 K/min	25 K/min	50 K/min	100 K/min	Unit
**Temperature**							
Melting *T_m_*	217.5	218.3	220.1	227.6	229.8	230.2	°C
Crystallization *T_Cr_*	140.4	130.6	127.1	165.3	150.7	140.2	°C
Glass Transition *T_g_*	55.1	45.4	44.6	65.6	63.4	62.4	°C
**Enthalpy**							
Melting Δ*H_m_*	−38.4	−37.3	−36.8	−39.1	−38.5	−37.9	J/g
Crystallization Δ*H_Cr_*	−59.5	−58.7	−57.6	−59.8	−58.8	−57.9	J/g

**Table 4 polymers-13-02401-t004:** Specimen qualifications made with PA6.

	P1	P2	P3	P1c	P2c	P3c	P1s	P2s	P3s	
Property		Value			Value			Value		Unit
**Geometry**										
Height *H*	20.00	19.90	20.00	20.00	19.90	19.95	20.00	20.00	19.90	mm
Diameter *D*	19.90	19.92	19.94	19.85	19.85	19.78	19.90	19.95	19.97	mm
Volume *V*	6221	6202	6246	6189	6158	6130	6221	6252	6233	mm^3^
**Density**										
Weight *m*_1_	7.08	7.02	7.09	7.01	6.98	6.94	7.09	7.11	7.10	g
Density 𝜌	1.14	1.13	1.14	1.13	1.13	1.13	1.14	1.14	1.14	g/cm^3^
Error *e*	0.16	0.71	0.42	0.65	0.58	0.70	0.02	0.24	0.08	%

**Table 5 polymers-13-02401-t005:** Specimen qualifications made with PA6GF.

	P1	P2	P3	P1c	P2c	P3c	P1s	P2s	P3s	
Property		Value			Value			Value		Unit
**Geometry**										
Height *H*	20.00	19.95	20.00	19.85	19.90	19.95	20.00	19.95	20.00	mm
Diameter *D*	19.80	19.72	19.72	20.00	19.90	19.82	19.80	19.82	19.80	mm
Volume *V*	6158	6093	6108	6236	6189	6155	6158	6155	6158	mm^3^
**Density**										
Weight *m*_1_	7.19	7.11	7.13	7.29	7.22	7.14	7.15	7.12	7.18	g
Density 𝜌	1.17	1.17	1.17	1.17	1.17	1.16	1.16	1.16	1.17	g/cm^3^
Error *e*	0.21	0.27	0.24	0.08	0.30	0.85	0.76	1.13	0.35	%

**Table 6 polymers-13-02401-t006:** Results of the compression tests on dry-as-mold specimens.

	PA			PA6GF			
Property	P1	P2	P3	P1	P2	P3	Unit
Modulus *E_c_*	828.56	777.39	780.08	857.04	812.49	838.45	MPa
Yield stress *σ_10_*/*σ_y_*	76.78	74.71	80.25	72.25	73.94	67.55	MPa
Maximum stress *σ_m_*	92.76	93.01	91.61	144.07	146.90	145.09	MPa

**Table 7 polymers-13-02401-t007:** Results of compression tests on dry, conditioned (con), and saturated (sat) specimens.

	PA			PA6GF			
Property	Dry	Con	Sat	Dry	Con	Sat	Unit
Modulus *E_c_*	795.34	772.35	754.51	835.99	492.90	261.35	MPa
Yield stress *σ*_10_/*σ_y_*	77.24	70.30	69.21	72.91	48.60	25.44	MPa
Maximum stress *σ_m_*	92.46	88.56	88.55	145.35	108.12	55.71	MPa

## Data Availability

The data presented in this study are available on request from the corresponding author.

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
