# Peer review of "Hygromechanical Performance of Polyamide Specimens Made with Fused Filament Fabrication"

_polymers, 2021, doi:10.3390/polym13152401_

Round 1

Reviewer 1 Report

The authors have applied commercial filaments in this study, so DSC and FTIR analysis on the standard filaments needed to be verified. Please highlight what is the purpose on doing the analysis in this study. Is there any helps on further discussion or experimental works.  

100% infills show best performances with dimensional & volume changes and moisture absorption were found expected negligible in this study.

However, I don’t think 100% infills will be chosen when product fabrication, because this will made production extremely slow, heavier (than <100% infills). This will prompt the producers to fabricate with other processing methods, like injection moulding. Besides, the results show filament without glass fibres specimens having better dimensional integrity. However, this may be different if the infills was less than 100%.

Hence, I found there are little interests on 100% infill specimens. The authors should clarify the printing parameters.

Surface morphologies should add in this study, to check the surface conditions and bonding conditions between the layers. This might be the factor of mechanical properties determiners in FFF fabrications.

Author Response

Thank you to the reviewer for the precious suggestions in correcting the paper. We made the necessary modifications.

Q1. The authors have applied commercial filaments in this study, so DSC and FTIR analysis on the standard filaments needed to be verified. Please highlight what is the purpose on doing the analysis in this study. Is there any helps on further discussion or experimental works.

R1. The reasons for conducting DSC and FT-IR analysis were to identify the correct process parameters linked to the thermal properties and to assess the influence of the environment on filament degradation.

These sentences were added to the text (lines 248-255 and lines 267-272).

Q2. 100% infills show best performances with dimensional & volume changes and moisture absorption were found expected negligible in this study. However, I don’t think 100% infills will be chosen when product fabrication, because this will made production extremely slow, heavier (than <100% infills). This will prompt the producers to fabricate with other processing methods, like injection moulding. Besides, the results show filament without glass fibers specimens having better dimensional integrity. However, this may be different if the infills was less than 100%. Hence, I found there are little interests on 100% infill specimens. The authors should clarify the printing parameters.

R2. The choice of a 100% infill was fundamental to achieving the highest value for the Young’s Modulus, avoiding the difference induced by selecting the infill pattern. The variation of the infill percentage and pattern will be a topic for future research.

These sentences were added to the text (lines 175-177 and lines 420-422).

Q3. Surface morphologies should add in this study, to check the surface conditions and bonding conditions between the layers. This might be the factor of mechanical properties determiners in FFF fabrications.

R3. The analysis of the specimens sections with the optical microscopy is reported in the text.

The sentences were added to the text (lines 220-227 and lines 393-403).

Reviewer 2 Report

Authors described in detail the changes in the FFF formulations with moisture. I think it is a very descriptive manuscript and the aim is clear. 

I just want reccommend authors to do not use abreviations in the abstract (as PA), can be confusing for readers. Also, I miss some more results in the abstract to fully understand the content. 

I suggest the editor to accept this manuscript after minor corrections. 

Author Response

Thank you to the reviewer for the precious suggestions in correcting the paper. We made the necessary modifications.

Q1. I just want recommend authors to do not use abbreviations in the abstract (as PA), can be confusing for readers. Also, I miss some more results in the abstract to fully understand the content. .

R1. The abstract was modified and abbreviation explained.